# Capacity of Retinal Ganglion Cells Derived from Human Induced Pluripotent Stem Cells to Suppress T-Cells

**DOI:** 10.3390/ijms21217831

**Published:** 2020-10-22

**Authors:** Ayaka Edo, Sunao Sugita, Yoko Futatsugi, Junki Sho, Akishi Onishi, Yoshiaki Kiuchi, Masayo Takahashi

**Affiliations:** 1Laboratory for Retinal Regeneration, RIKEN Center for Biosystems Dynamics Research, 2-2-3 Minatojima-minamimachi, Chuo-ku, Kobe 650-0047, Japan; ayakae@hiroshima-u.ac.jp (A.E.); yoko.futatsugi@riken.jp (Y.F.); junki.sho@riken.jp (J.S.); akishi.onishi@riken.jp (A.O.); retinalab@ml.riken.jp (M.T.); 2Department of Ophthalmology and Visual Science, Graduate School of Biomedical and Health Sciences, Hiroshima University, 1-2-3 Kasumi, Minami-ku, Hiroshima 734-8551, Japan; ykiuchi@hiroshima-u.ac.jp

**Keywords:** retinal ganglion cells, induced pluripotent stem cells, immunogenicity, mixed lymphocyte reaction, immunosuppression

## Abstract

Retinal ganglion cells (RGCs) are impaired in patients such as those with glaucoma and optic neuritis, resulting in permanent vision loss. To restore visual function, development of RGC transplantation therapy is now underway. Induced pluripotent stem cells (iPSCs) are an important source of RGCs for human allogeneic transplantation. We therefore analyzed the immunological characteristics of iPSC-derived RGCs (iPSC-RGCs) to evaluate the possibility of rejection after RGC transplantation. We first assessed the expression of human leukocyte antigen (HLA) molecules on iPSC-RGCs using immunostaining, and then evaluated the effects of iPSC-RGCs to activate lymphocytes using the mixed lymphocyte reaction (MLR) and iPSC-RGC co-cultures. We observed low expression of HLA class I and no expression of HLA class II molecules on iPSC-RGCs. We also found that iPSC-RGCs strongly suppressed various inflammatory immune cells including activated T-cells in the MLR assay and that transforming growth factor-β2 produced by iPSC-RGCs played a critical role in suppression of inflammatory cells in vitro. Our data suggest that iPSC-RGCs have low immunogenicity, and immunosuppressive capacity on lymphocytes. Our study will contribute to predicting immune attacks after RGC transplantation.

## 1. Introduction

Retinal ganglion cells (RGCs), which are located in the innermost layer of the retina, are responsible for collecting optical information that reaches the retina, and transmitting it to the brain via the optic nerve [1]. Retinal pigment epithelial cells (RPEs), which are one of the constituent cells of the retina as well as RGCs, have been reported to possess immune-suppression capacity [2,3,4,5,6]. Stem cell-derived RPE transplantation therapy has already been applied clinically [7,8,9,10]. On the other hand, to date, no reports have described the immunological characteristics of RGCs. We hypothesized that RGCs may also have the immunosuppressive properties such as RPEs. Degeneration and apoptosis of RGCs that occur in glaucoma and optic neuropathy can cause irreversible visual impairment [11,12,13,14]. In particular, glaucoma is a type of progressive degeneration of RGCs that is estimated to affect more than 70 million people worldwide and is the leading cause of irreversible blindness [12,15,16]. Like other neurons of the central nervous system, RGCs do not readily self-replicate, and are thus unrecoverable once they are impaired in mammals including humans [17]. No treatment is available to restore vision lost due to RGC-related disorders, and development of cell replacement therapies is urgently needed. A better approach to restore vision is focused on functional replacement of RGCs by transplantation such as autologous grafts or allografts using stem cells, and is being studied all over the world [14,18]. Recent studies have reported that in vivo transplanted RGCs in animal models are engrafted into the host retina and respond to light stimulation [17,18,19,20,21]. Cell replacement therapy for RGCs has advanced by leaps and bounds.

Because auto-transplantation, especially induced pluripotent stem cell (iPSC)-derived cells/tissues, is very costly and time-consuming for cell preparation, allogeneic transplantation is a practical way to promote standard treatment. Especially in allogenic transplantation, rejection is a critical concern. In general, CD4-positive and CD8-positive T-cells are largely responsible for rejection [22,23,24]. Expression of human leukocyte antigen (HLA) and CD80/CD86 co-stimulatory molecules is a major trigger of immune response, and high expression these molecules increases the risk of rejection [25,26,27]. T-cell suppressive capacity and low expression of HLA molecules are associated with the low incidence of rejection. Characterization of RGC immunology is important for assessing the possibility of immune rejection after RGC transplantation.

In the present study, we evaluated the immunogenicity of iPSC-RGCs such as expression of HLA molecules, and CD80, CD86, and CD274 co-stimulatory molecules. We also investigated whether iPSC-RGCs have a T-cell suppression capacity using the mixed lymphocyte reaction (MLR) assay, which is extensively used for assessing immune rejection in vitro [28,29,30,31,32]. To elucidate the immunosuppressive property and the immunogenicity of human iPSC-derived RGCs (iPSC-RGCs) against lymphocytes, we used an in vitro model with iPSC-RGCs derived from healthy donors co-cultured with active lymphocytes isolated from the peripheral blood of healthy subjects.

## 2. Results

### 2.1. Preparation of iPSC-RGCs

First, we established human iPSCs (TLHD2 line) from peripheral blood mononuclear cells (PBMCs) derived from a healthy donor. The morphology of stem cell colonies was well formed (Figure 1A). iPSCs expressed the pluripotency markers, Nanog and SSEA4, as seen with immunocytochemistry (ICC), whereas no staining was seen with the rabbit isotype control (Figure 1B). Karyotype analysis of iPSCs showed no chromosomal aberrations affecting the phenotype and only an inversion of chromosome 9, which is considered to be a structural chromosomal polymorphism in the general population (Figure 1C) [33,34].

We then differentiated iPSCs into 3D retinal tissue to collect iPSC-RGCs (Figure 1D). In the 3D retina at differentiation day (DD) 50, the RGC marker, Brn-3b, was clearly expressed, and Crx, which is a photoreceptor marker, was also expressed as seen with immunohistochemistry (IHC) (Figure 1E).

iPSC-RGCs were successfully purified from 3D retina with a previously described method [17,35,36], and the purified iPSC-RGCs showed dendrite-like morphology (Figure 2A). They clearly expressed RGC markers such as Brn-3b and axonal neurofilament markers such as SMI-312 in ICC (Figure 2B). iPSCs were not stained with Brn-3b and SMI-312 (Appendix A). *Brn-3b*, *RBPMS*, *ISL1*, and *THY1*, which are RGC-related genes, were highly expressed in iPSC-RGCs compared to control PBMCs (the same donor) as seen with quantitative reverse transcription-polymerase chain reaction (RT-PCR) analysis (Figure 2C).

### 2.2. Immunogenicity of iPSC-RGCs Assessed with HLA Class I, Class II, and Co-Stimulatory Molecules

To examine the immunogenicity of iPSC-RGCs, we first analyzed HLA class I (HLA-A, B, C) and class II (HLA-DR, DP, DQ) molecules on iPSC-RGCs with ICC. We used iPSC-derived retinal pigment epithelial cells (iPSC-RPEs: same donor, TLHD2) as a positive control. The iPSC-RGCs poorly expressed HLA class I molecules unlike iPSC-RPEs (Figure 3A). In the presence of interferon-γ (IFN-γ), which is a cytokine that induces the inflammatory state in vitro [6], the expression of HLA class I increased in iPSC-RGCs as well as in iPSC-RPEs. However, HLA class I on RGCs stimulated with IFN-γ was lower than that on similarly stimulated iPSC-RPEs (Figure 3A). On the other hand, HLA class II molecules were not expressed on iPSC-RGCs or on iPSC-RPEs. No expression of HLA class II molecules on IFN-γ-treated iPSC-RGCs was observed despite the weak expression of HLA class II on IFN-γ-treated iPSC-RPEs (Figure 3A).

We next performed flow cytometric analysis and examined the expression of the co-stimulatory molecules, CD80 (B7-1) and CD86 (B7-2), which activate T-cells, and CD274 (PD-L1: B7-H1), which inhibits proliferation and cytokine production by activated T-cells [37,38]. CD80 and CD86 co-stimulatory molecules were not expressed on iPSC-RGCs in the presence or absence of IFN-γ. CD274 was slightly expressed on iPSC-RGCs, and its expression was enhanced by IFN-γ pre-treatment (Figure 3B). The expression of these co-stimulatory molecules on PBMCs (positive control: PC) was confirmed.

In summary, iPSC-RGCs express HLA class I, although poorly, and the expression was enhanced in inflammatory conditions. HLA class II, CD80, and CD86 were not expressed on iPSC-RGCs regardless of the presence or absence of inflammatory conditions. CD274, which inhibits T-cell activation, was weakly expressed on iPSC-RGCs, and was enhanced in inflammatory conditions. Our results demonstrate that the immunogenicity of iPSC-RGCs is very low. We next confirmed whether iPSC-RGCs could suppress inflammatory immune cells in vitro.

### 2.3. Human iPSC-RGCs are able to Suppress Immune Cell Activation In Vitro

We next performed the MLR assay to examine whether iPSC-RGCs inhibit activated immune cells. The MLR assay is widely used to evaluate allogeneic immune responses in vitro [28,29,30,31,32]. We used iPSCs as a control (TLHD2 line). In this assay, the activation of PBMCs was analyzed with Ki-67 proliferation fluorescence-activated cell sorting (FACS) analysis and IFN-γ secretion as detected with an enzyme-linked immunosorbent assay (ELISA). Figure 4A and Appendix A shows the results of Ki-67 proliferation FACS analysis in the MLR assay (lymphocyte proliferation). We found that the proliferation rates of CD3-positive pan T-cells, CD4-positive helper T-cells, CD8-positive cytotoxic T-cells, CD11b-positive monocytes and macrophages, and CD159a-positive natural killer (NK) cells that were co-cultured with iPSC-RGCs were significantly lower compared to those cells co-cultured with iPSCs. iPSC-RGCs greatly suppressed various inflammatory immune cells in the MLR assay (Figure 4A,B). In contrast, proliferation rates of these immune cells co-cultured with iPSCs were not significantly different compared to those of the MLR cell monoculture. iPSCs did not suppress lymphocytes in the MLR assay (Figure 4A,C). We then measured the concentration of IFN-γ in the culture supernatant with ELISA. The concentration of IFN-γ in the supernatant containing MLR-RGC co-cultures was significantly lower than that of MLR-iPSC co-cultures (Figure 4D). IFN-γ is produced by CD4-positive Th1 effector T-cells, CD8-positive cytotoxic T-cells, and NK cells [39]. This result corroborates the results of the Ki-67 proliferation FACS analysis.

Even in the condition of induced T-cell activation following stimulation with anti-human CD3 agonistic antibody, iPSC-RGCs also inhibited the proliferation of CD3-positive pan T-cells, CD4-positive helper T-cells, and CD8-positive cytotoxic T-cells (Appendix A). These results indicated that iPSC-RGCs have the capacity to inhibit activation of T-cells in vitro.

We also confirmed whether the ratio of iPSC-RGCs to MLR cells affects the immunosuppressive effects of RGCs. We co-cultured 5 × 10^5^ PBMCs in the MLR assay with 5.0 × 10^3^ to 2.5 × 10^5^ iPSC-RGCs (MLR:RGC ratio = 100:1, 50:1, 10:1, 5:1, and 2:1). After 120 h of incubation, Ki-67 proliferation FACS analysis was performed. We observed that 2.5 × 10^5^ cells and 1.0 × 10^5^ iPSC-RGCs (MLR:RGC ratio = 2:1 and 5:1) inhibited proliferation of CD4-positive and CD8-positive T-cells in vitro. On the other hand, 5.0 × 10^3^ to 5.0 × 10^4^ iPSC-RGCs (MLR:RGC ratio = 100:1, 50:1, and 10:1) failed to suppress T-cell proliferation (Appendix A).

### 2.4. Elucidating of the Immunosuppressive Mechanism of iPSC-RGCs

To identify the immunosuppressive mechanism of iPSC-RGCs, we surveyed expression of immunomodulatory molecules/factors in iPSC-RGCs using DNA microarray analysis. iPSCs and PBMCs (both from same donor: TLHD2) were prepared as controls. The results of microarray analysis are shown in Table 1. Among the T-cell-associated immunosuppressive factors, iPSC-RGCs highly expressed transforming growth factor beta 2 (TGF-β2), thrombospondin-1 (TSP-1), and somatostatin compared to control cells. In microarray analysis, the expression of HLA class I and class II in iPSC-RGCs was lower compared to control cells. Co-stimulatory molecules were not expressed as highly as TGF-β2, TSP-1, and somatostatin. We therefore considered these three molecules, TGF-β2, TSP-1, and somatostatin, as candidate molecules related to the immunosuppressive effect of iPSC-RGCs.

To confirm the above results, we investigated the immunosuppressive effect of these candidate molecules on MLR cells by measuring IFN-γ with ELISA. As shown in Figure 5, recombinant human TGF-β2 (0.5 ng/mL, 5 ng/mL, 50 ng/mL) significantly suppressed IFN-γ production in MLR cells, but recombinant TSP-1 and somatostatin did not (Figure 5). These results suggest that TGF-β2 may play an important role in iPSC-RGC-mediated T-cell suppression.

### 2.5. Detection of TGF-β in iPSC-RGCs

To confirm the expression of TGF-β2 in iPSC-RGCs, quantitative RT-PCR and immunostaining were performed. Quantitative RT-PCR analysis showed that iPSC-RGCs highly expressed TGF-β2 and -β3 mRNA, and they expressed TGF-β1 mRNA to a similar extent as control cells (Figure 6A). ICC showed that iPSC-RGCs clearly expressed TGF-β2 on their surface (Figure 6B). iPSCs did not express TGF-β2 as seen with ICC (Appendix A). Based on these findings, we focused on RGC-derived TGF-β as an immunoregulatory factor that suppressed activated T-cells.

### 2.6. Role of TGF-β in T-Cell Suppression by iPSC-RGCs

To investigate the involvement of TGF-β in iPSC-RGC-induced inhibition of T-cells, we analyzed T-cell activation by inhibiting TGF-β in co-cultures of MLR cells and iPSC-RGCs. We inhibited TGF-β signaling using SB431542, a TGF-β receptor I inhibitor [6,40]. Ki-67 FACS proliferation analysis showed that the percentage of proliferating CD4-positive and CD8-positive cells that were co-cultured with iPSC-RGCs in the presence of SB431542 increased compared to those without SB431542 (Figure 7A). The same experiment was performed three times, and similar results were obtained (Appendix A). Secretion of IFN-γ from lymphocytes in MLR cells co-cultured with RGCs in the presence of SB431542 was partially but significantly increased compared with co-culture in the absence of SB431542 (Figure 7B). These results suggest that iPSC-RGCs suppress T-cell activation by TGF-β.

## 3. Discussion

In this study, we found that iPSC-RGCs have poor expression of HLA class I and no expression of HLA class II, CD80, and CD86 co-stimulatory molecules. When HLA class I of transplanted cells is recognized by immune cells, CD8-positive cytotoxic T-cells are activated to eliminate the grafts [22,23,24]. HLA class II, CD80, and CD86 co-stimulatory molecules are expressed on the antigen-presenting cells [25,31,38]. Our findings imply that iPSC-RGCs are less likely to activate CD8-positive cytotoxic T-cells and do not have an antigen-presenting function.

iPSC-RGCs expressed CD274 co-stimulatory molecules, which is a ligand of PD-1 expressed on T-cells [23]. PD-1 pathway inhibits the activation of CD4-positive and CD8-positive T-cells in vitro [41]. In allogenic heart transplants in mice in vivo, graft survival was prolonged by promoting PD-1 pathway [42]. This suggests that CD274 expression on iPSC-RGCs might have an advantage in preventing rejection.

We showed that iPSC-RGCs suppressed the immune activity of T-cells via TGF-β. Human corneal endothelial cells, iris pigment epithelial cells and stem cell-derived RPEs have been reported to suppress T-cell activation via TGF-β [4,6,43,44]. In particular, it has been suggested that the induction of regulatory T-cells by TGF-β is involved in the mechanism of inhibition by human corneal endothelial cells and iPSC-RPEs [4,44]. In the present study, the involvement of TGF-β has been suggested and a similar mechanism can be inferred.

SB431542, which was used to inhibit TGF-β signaling, did not completely block the inhibition of T-cell activation by iPSC-RGCs. In human RPEs, some factors other than TGF-β, such as prostaglandin E2 and soluble CD54, have also been reported to be involved in suppression of T-cell activity [3]. Thus, TGF-β mainly contributes to the T-cell suppression by iPSC-RGCs, but other immunosuppressive molecules other than TGF-β may also contribute.

TGF-β, TSP-1, and somatostatin were identified with DNA microarray analysis as candidate molecules involved in the immunosuppressive effects of RGCs. In the current study, TGF-β was largely involved in the suppression of T lymphocytes, whereas neither TSP-1 nor somatostatin directly inhibited the activation of lymphocytes (Figure 5). TSP-1 and somatostatin are immunosuppressive factors in the aqueous humor and vitreous humor that activate TGF-β [45,46,47,48,49,50,51]. Therefore, these two factors may have indirectly assisted the immunosuppression by TGF-β by activating the action of TGF-β by RGCs.

Based on our results mentioned above, we expect that immune rejection after iPSC-RGC transplantation will be weak or none, if it occurs at all, for the following reasons. First, iPSC-RGCs have an inhibitory effect on CD4-positive and CD8-positive T-cells. The activation of CD4-positive and CD8-positive T-cells is largely responsible for the formation of rejection. Cytotoxicity generated by CD8-positive T-cells upon recognition of HLA class I expressed on transplanted cells has been reported to be the main cause of rejection. CD4-positive helper T-cells activated by antigen presentation via HLA class II also promote rejection [22,23,24]. Second, the vitreous, into which iPSC-RGCs are injected at the time of transplantation, is an environment that is unlikely to cause inflammation. The vitreous contains a variety of anti-inflammatory factors such as TGF-β, vasoactive intestinal peptide, alpha-melanocyte stimulating hormone (α-MSH), somatostatin, and substance P [50]. Moreover, the vitreous has a mechanism to induce systemic immune tolerance by recognizing antigens and raising regulatory T-cells with anti-inflammatory effects. When antigens are injected into the vitreous, antigen-presenting cells in the vitreous or in the retina migrate to the spleen for antigen presentation, and activate antigen-specific regulatory T-cells to suppress the immune response in the vitreous [50,52,53].

Immune rejection occurs after allogeneic transplantation of iPSC-RPEs in vivo, despite the fact that the transplant iPSC-RPEs are capable of inhibiting T-cell activity and the transplantation site, the sub-retinal space, is an environment less prone to inflammation as well as the vitreous [54,55,56]. We also reported that immune rejection occurs in one of five age-related macular degeneration patients after allogeneic transplantation of iPSC-RPEs [10]. In humans, however, we would expect that iPSC-RGCs would be less likely to cause rejection than transplantation of iPSC-RPEs. The reason for this is that the expression of HLA molecules on RGCs is considerably lower than in iPSC-RPEs. Furthermore, transplantation of iPSC-RGCs may be less invasive than the iPSC-RPE transplantation technique. Specifically, transplantation of iPSC-RPEs requires a needle to be injected into the sub-retina [56,57], whereas transplantation of iPSC-RGCs requires only an injection into the vitreous cavity [17,18,19,20,21]. We believe that a minimally invasive procedure would reduce the risk of postoperative inflammation and easily prevent the enhancement of immunogenicity. This is also because iPSC-RGC transplantation is characterized by an unbroken blood retinal barrier and less systemic blood flow into the eye if the host RPE is not compromised or if no retinal hemorrhage occurs [49].

In animal models of RGC transplantation, in which mouse RGCs were transplanted into rat vitreous, transplanted cells survived and no obvious signs of rejection were observed at 3 weeks after transplantation without the use of immunosuppressive drugs [17]. This suggests that the risk of rejection associated with RGC transplantation is relatively low.

The next step to predict rejection based on the results of this study would be to evaluate the immune rejection in animal transplants using iPSC-RGCs. We hope that our study will support the development of strategies against rejection for RGC transplantation therapy that is becoming nearing reality.

## 4. Materials and Methods

### 4.1. Establishment of Human iPSCs

This study followed the tenets of the Declaration of Helsinki, and was approved by the Institutional Ethics Committee of the RIKEN BDR (Approval No. Kobe1 2013-01(10), 29 March 2019). After informed consent was obtained from a donor (TLHD2), iPSCs were established from PBMCs using an episomal vector [58] by iPS Portal (Kyoto, Japan). The karyotyping of the iPSCs was performed by Japan Gene Research Laboratories (Miyagi, Japan).

### 4.2. Preparation, Isolation, and Culture of iPSC-RGCs

Feeder-free iPSCs from healthy donors (TLHD2 and 201B7 lines [59,60]) were differentiated into 3D retina as previously described [36,60]. 201B7 iPSCs were kindly provided from the Center for iPS Cell Research and Application, Kyoto University. iPSC-RGCs were purified from 3D retina at DD 55-65 using a two-step immunopanning method as previously described [17,35,36]. RGCs isolated using immunopanning were more than 99.5% pure [35]. The isolated iPSC-RGCs were suspended in RGC medium and seeded in a 96-well plate coated with poly-d-lysine (Sigma-Aldrich, St. Louis, MO, USA) and laminin (Sigma-Aldrich). RGC medium was composed of Neurobasal medium (Invitrogen Life Sciences, Carlsbad, CA, USA) with brain-derived neurotrophic factor (50 ng/mL; PeproTech, Rocky Hill, NJ, USA), ciliary neurotrophic factor (50 ng/mL; PeproTech), basic fibroblast growth factor (50 ng/mL; PeproTech), forskolin (10 mM; Sigma-Aldrich), B27 supplement (2%; Invitrogen Life Sciences), l-glutamine (1 mM; Thermo Fisher Scientific, Waltham, MA, USA), insulin (5 µg/mL; Sigma-Aldrich), sodium pyruvate (1 mM; Thermo Fisher Scientific), progesterone (62 ng/mL; Sigma-Aldrich), putrescine (16 µg/mL; Sigma-Aldrich), sodium selenite (40 ng/mL; Sigma-Aldrich), triiodothyronine (40 ng/mL; Sigma-Aldrich), and N-acetylcysteine (5 µg/mL; Sigma-Aldrich). iPSC-RGCs at 3-5 days after isolation were used for the experiments. For some assays, iPSC-RGCs were pre-treated with IFN-γ (100 ng/mL; R&D Systems, Minneapolis, MN, USA) for 48-96 h. iPSC-RPEs as a control were prepared as previously described [4].

### 4.3. Flow Cytometry

The expression of CD80 (B7-1), CD86 (B7-2), and CD274 (PD-L1/B7-H1) on iPSC-RGCs was assessed with FACS analysis. Cells were washed with RPMI 1640 (Nacalai Tesque, Kyoto, Japan) with 2% fetal bovine serum (Thermo Fisher Scientific) and incubated with blocking human Fc (Miltenyi Biotec, Bergisch Gladbach, Germany) on ice for 10 min. Then, the cells were stained with FITC-labeled anti-human CD80, FITC-labeled anti-human CD86, PE-labeled anti-human CD274, or the appropriate isotype-matched controls on ice for 30 min in the dark.

Ki-67 staining in FACS analysis was performed with the following procedure for intracellular staining for Ki-67. PBMCs were fixed and permeabilized in 70% ethanol at −30°C for 1 h and then washed three times using cell staining buffer (BioLegend, San Diego, CA, USA). They were stained with PE-labeled anti-Ki-67 antibody, APC-labeled anti-CD3 antibody, APC-labeled CD4 antibody, APC-labeled CD8 antibody, APC-labeled CD11b antibody, or APC-labeled CD159a (NKG2A) antibody for 40 min at 25 °C. All antibody information is provided in Appendix A. Flow cytometric analysis was performed using the BD Canto II flow cytometer (BD Biosciences, San Jose, CA, USA), and the obtained data were analyzed with FlowJo software (Tree Star, Ashland, OR, USA).

### 4.4. Quantitative RT-PCR

Expression of mRNA levels of *Brn-3b*, *ISL1*, *RBPMS*, *THY1*, *TGF-β1*, *-β2*, *-β3*, and *CD3* were evaluated using quantitative RT-PCR. Total RNA from iPSC-RGCs and control PBMCs from the TLHD2 donor was purified using the High Pure RNA Isolation Kit (Roche, Basel, Switzerland). After cDNA synthesis using Transcriptor First Strand cDNA Synthesis Kit (Roche), the expression of various molecules and glyceraldehyde 3-phosphate dehydrogenase (*GAPDH*) in triplicate samples was analyzed with quantitative RT-PCR (LightCycler model 480; Roche), as previously described [4,36]. The results show the relative expression of the target molecules (delta cycle threshold (∆∆Ct): control cells = 1). The sequences of the primers and probes are described in Appendix A.

### 4.5. IHC and ICC

The iPSC markers, Nanog and SSEA-4, on iPSCs were confirmed with ICC as previously described [4]. IHC for Brn-3b and Crx in 3D retina and ICC for HLA class I, class II, Brn-3b, SMI-312, and TGF-β2 in iPSC-RGCs were performed as described previously [36]. Nuclei were counterstained with 4′,6-diamidino-2-phenylindole (DAPI: Invitrogen). The specimens were visualized with a confocal microscope (LSM700; Carl Zeiss, Jena, Germany). All antibodies used in IHC and ICC are described in Appendix A.

### 4.6. MLR Assay with iPSC-RGCs

PBMCs were isolated from healthy donors using Lymphoprep (Stemcell Technologies, Vancouver, Canada) according to the manufacturer’s instructions. Allogeneic immune responses were assessed for the percentage of PBMCs expressing Ki-67 (proliferation marker) using Ki-67 FACS proliferation analysis, and the concentration of IFN-γ secreted from lymphocytes in PBMCs was measured using IFN-γ ELISA. PBMCs were cultured in RPMI 1640 (Nacalai Tesque) with 10% fetal bovine serum, human recombinant interleukin-2 (BD Biosciences), 10 mM HEPES (Sigma-Aldrich), 1% nonessential amino acids (Sigma-Aldrich), 1% sodium pyruvate (Sigma-Aldrich), 1 × 10^−5^ M 2-mercaptoethanol (Sigma-Aldrich), and 1% penicillin/streptomycin (Thermo Fisher Scientific). Mixed PBMCs (5 × 10^5^ cells/well in 96-well plates) from five healthy donors were co-cultured with iPSC-RGCs (2.5 × 10^5^ cells/well). After 120 h of incubation, PBMCs were analyzed with Ki-67 FACS, and the concentration of IFN-γ in the supernatants was measured using the Human IFN-γ ELISA Kit (R&D Systems). We used iPSCs as a control. To investigate whether the ratio of iPSC-RGCs to MLR cells affects the immunosuppressive effect of RGCs, 5 × 10^5^ MLR cells and 5.0 × 10^3^ to 2.5 × 10^5^ iPSC-RGCs were co-cultured and analyzed with Ki-67 FACS.

To analyze the immune responses by inhibiting TGF-β signaling, we used a TGF-β receptor I inhibitor (SB431542; Sigma-Aldrich). SB431542 (10 μM) was added to a co-culture of MLR cells and iPSC-RGCs. After 120 h of incubation, T-cell activation was measured with Ki-67 FACS and IFN-γ secretion.

To compare the immunosuppressive effects of candidate molecules involved in RGC-induced immunosuppression, we added human recombinant proteins such as TGF-β2, TSP-1, and somatostatin (0.5, 5, 50 ng/mL: all from R&D Systems) to MLR cells. After 120 h of incubation, IFN-γ secretion in the supernatant was measured.

### 4.7. T-cell Proliferation Assay with iPSC-RGCs

To evaluate the immunosuppressive effects of iPSC-RGCs on T-cell proliferation in PBMCs induced by human CD3 agonistic antibody (Ancell, Bayport, MN, USA), PBMCs and iPSC-RGCs were co-cultured. PBMCs from the healthy donor were incubated with iPSC-RGCs in the presence of 0.1 µg/mL human CD3 antibody for 72 h, and PBMCs were analyzed with Ki-67 FACS.

### 4.8. Expression Analysis with Microarray

DNA microarray was performed to reveal the gene expression of iPSC-RGCs. iPSCs and PBMCs (from the TLHD2 donor) were used as control cells. RNA from iPSC-RGCs, iPSCs, and PBMCs was extracted using the High Pure RNA Isolation Kit (Roche). The Clariom™ S Assay, human (Thermo Fisher Scientific) was performed by Filgen (Aichi, Japan). RNA quantity and quality were measured with an Agilent 2100 Bioanalyzer (Agilent Technologies, Inc., Santa Clara, CA, USA). Preparation of labeled samples and hybridization were performed in accordance with the GeneChip™ WT PLUS Reagent Kit User Manual (Thermo Fisher Scientific). The array was scanned with the GeneChip™ Scanner 3000 7G (Thermo Fisher Scientific) in accordance with the GeneChip™ Command Console AGCC 4.0 User Manual (Thermo Fisher Scientific). We registered these data in the GEO database: Accession No. GSE154544.

### 4.9. Statistical Analysis

All in vitro experiments were repeated three times. Data are presented as mean ± standard deviation (SD) of the experimental triplicate. All statistical analysis was performed by Student’s *t*-test using JMP pro 14.0.0 (SAS Institute Inc., Cary, NC, USA). *p* < 0.05 was considered statistically significant.

## 5. Conclusions

We isolated iPSC-RGCs from human iPSC-derived 3D retina and analyzed the immune properties of iPSC-RGCs. We showed that iPSC-RGCs had low immunogenicity and a suppressive function on T-cells, and that TGF-β played a critical role in the suppressive mechanism. We report herein for the first time that RGCs are cells with an immunosuppressive capacity.

## Figures and Tables

**Figure 1 ijms-21-07831-f001:**
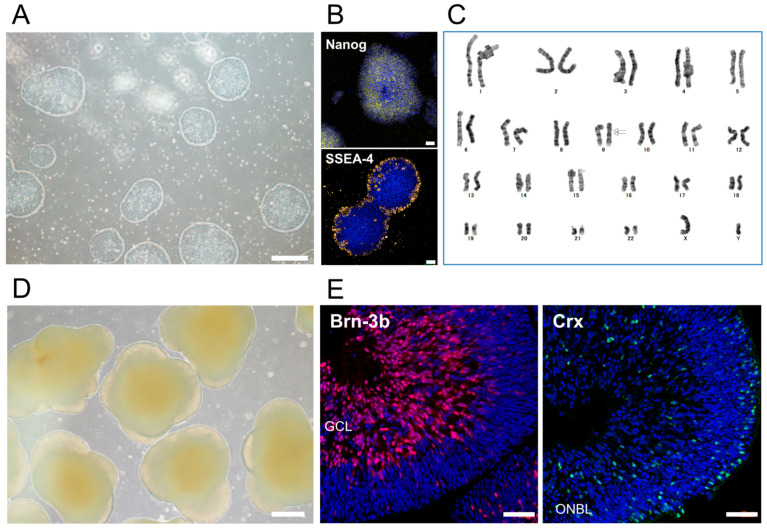
Established human iPSCs from a healthy donor and iPSC-derived three-dimensional retinal organoids (3D retina). (**A**) The morphology of feeder-free human iPSCs from a healthy donor. Circular colonies were well formed. Scale bar, 500 µm. (**B**) Expression of the pluripotency markers, Nanog (yellow) and SSEA4 (yellow), on iPSCs was detected with immunostaining. No staining was seen with the isotype control (rabbit IgG). Cell nuclei were counterstained with DAPI (blue). Scale bars, 50 µm. (**C**) The karyotype of iPSCs showed no chromosomal aberrations affecting the phenotype; only an inversion of chromosome 9 was seen. (**D**) The morphology of 3D retina at DD57 differentiated from iPSCs. Scale bar, 500 µm. (**E**) Expression of the RGC marker, Brn-3b, and photoreceptor marker, Crx, in 3D retina at DD50. 3D retina expressed Brn-3b (red) and Crx (green). Cell nuclei were counterstained with DAPI (blue). GCL, ganglion cell layer; ONBL, outer neuroblastic layer. Scale bars, 50 µm.

**Figure 2 ijms-21-07831-f002:**
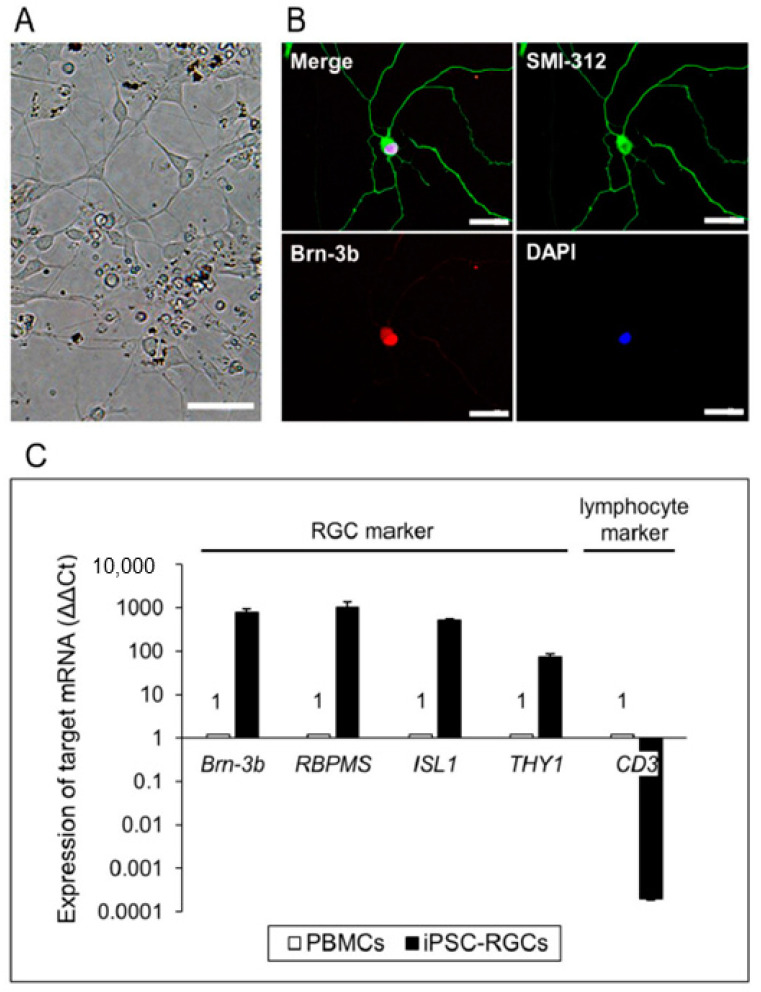
iPSC-derived RGCs (iPSC-RGCs) isolated from 3D retina using immunopanning. (**A**) The morphology of iPSC-RGCs at 3 days after isolation showed neurites with cell bodies (dendrite-like morphology). Scale bars, 50 µm. (**B**) Expression of the RGC marker, Brn-3b (red), and neurofilament marker, SMI-312 (green), on iPSC-RGCs was confirmed with ICC. No staining was seen with isotype controls (mouse, goat, and rabbit IgG). Cell nuclei were counterstained with DAPI. Scale bars, 50 µm. (**C**) Quantitative RT-PCR analysis of iPSC-RGCs. iPSC-RGCs (black bars) highly expressed mRNA for RGC-related genes (*Brn-3b*, *ISL1*, *RBPMS*, and *THY1*), whereas RGCs poorly expressed the lymphocyte marker, *CD3*. We used PBMCs (from the same TLHD2 donor: open bars) as control cells. Results show the relative expression (∆∆Ct: control cells = 1) and represent the mean ± standard deviation (SD) (*n* = 3).

**Figure 3 ijms-21-07831-f003:**
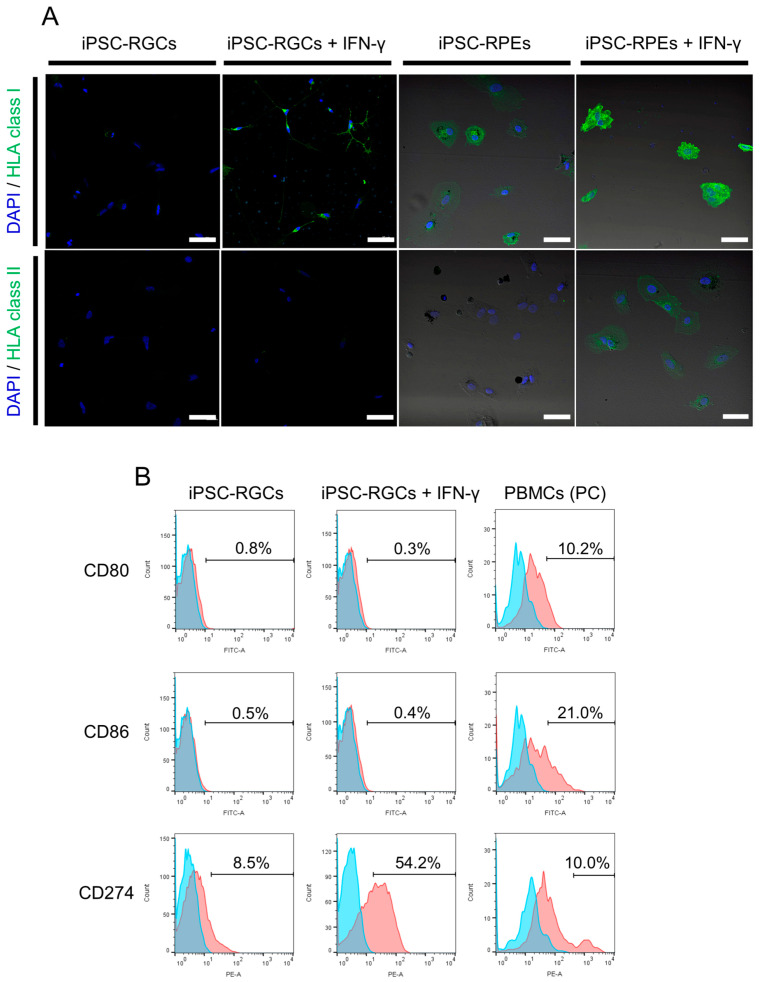
Expression of HLA class I, class II, and co-stimulatory molecules on human iPSC-RGCs. (**A**) Expression of HLA class I (green) and class II (green) on iPSC-RGCs was assessed with ICC. iPSC-RPEs were used as control cells. HLA class I was poorly expressed on iPSC-RGCs and was enhanced, but slightly, on IFN-γ-treated iPSC-RGCs. HLA class II was not expressed on iPSC-RGCs with or without IFN-γ pre-treatment. Cell nuclei were counterstained with DAPI (blue). Scale bars, 50 μm. (**B**) Expression of co-stimulatory molecules, CD80 (B7-1), CD86 (B7-2), and CD274 (PD-L1:B7-H1), was assessed with fluorescence-activated cell sorting analysis. CD80 and CD86 co-stimulatory molecules were not expressed on iPSC-RGCs with or without IFN-γ (red histograms). CD274 co-stimulatory molecules were slightly expressed on iPSC-RGCs and were enhanced on IFN-γ-treated iPSC-RGCs (red histograms). We also prepared PBMCs as positive control (PC) cells. Numbers in the histograms indicate the percentage of positive cells. Blue histograms represent the isotype control.

**Figure 4 ijms-21-07831-f004:**
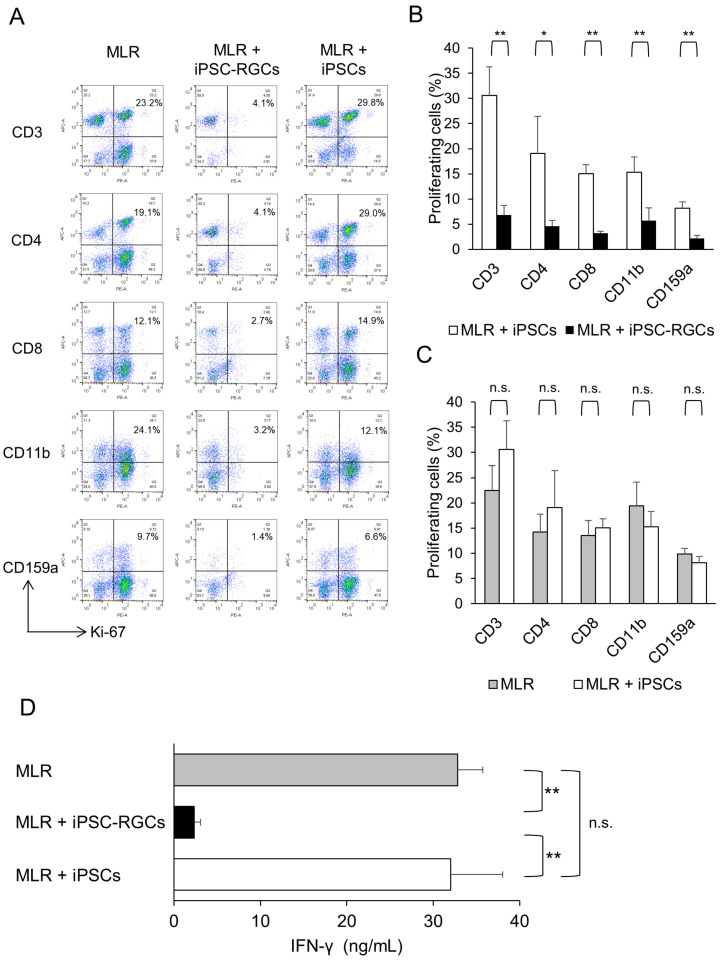
Capacity of iPSC-RGCs to suppress activated lymphocytes in the mixed lymphocyte reaction (MLR) assay. MLR cells (mixed PBMCs from five healthy donors) were co-cultured with iPSC-RGCs (MLR:RGC ratio = 2:1) for 120 h. To evaluate the capacity to suppress activation of immune cells by iPSC-RGCs, the activation of immune cells was analyzed with Ki-67 FACS, and IFN-γ secretion in the supernatants was measured with ELISA. We prepared iPSCs as control cells. (**A**) Representative data of Ki-67 FACS proliferation analysis. Numbers in the scatterplots indicate the percentage of double-positive cells for CD3, CD4, CD8, CD11b, or CD159a and Ki-67. The proliferation of CD3-, CD4-, CD8-, CD11b-, and CD159a-positive cells was greatly inhibited in the MLR assay when MLR cells were co-cultured with iPSC-RGCs. (**B**) The percentage of double-positive cells in Ki-67 FACS analysis compared between MLR cells plus iPSC-RGCs and MLR cells plus control iPSCs. The proliferation rates of CD3-, CD4-, CD8-, CD11b-, and CD159a-positive cells that were co-cultured with iPSC-RGCs were significantly lower compared to those co-cultured with iPSCs. The data represent the mean ± SD (*n* = 3). * *p* < 0.05, ** *p* < 0.01. (**C**) The percentage of double-positive cells in Ki-67 FACS analysis in the MLR cells plus iPSCs compared to that of MLR cells alone. No significant difference was found between the two groups. The data represent the mean ± SD (*n* = 3). *n.s*., not significant. (**D**) The concentration of IFN-γ in the supernatants of MLR cells plus iPSC-RGCs was significantly lower than that of MLR cells plus iPSCs. The concentration of IFN-γ in the supernatants of MLR cells plus iPSCs was not significantly different compared to that of MLR cells alone. The data represent the mean ± SD (*n* = 3). ** *p* < 0.01. *n.s*., not significant.

**Figure 5 ijms-21-07831-f005:**
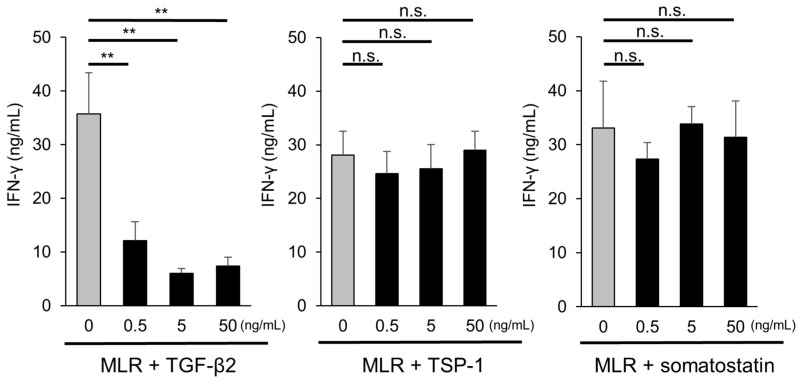
Capacity of candidate factors to suppress activation of T-cells. MLR cells were cultured with human recombinant TGF-β2, human recombinant TSP-1, and somatostatin (0.5, 5, 50 ng/mL), and the concentration of IFN-γ produced by T-cells was evaluated. Recombinant human TGF-β2 (0.5, 5, and 50 ng/mL, respectively) significantly suppressed IFN-γ production, although recombinant TSP-1 and somatostatin did not. The data represent the mean ± SD (*n* = 3). ** *p* < 0.01. *n.s*., not significant.

**Figure 6 ijms-21-07831-f006:**
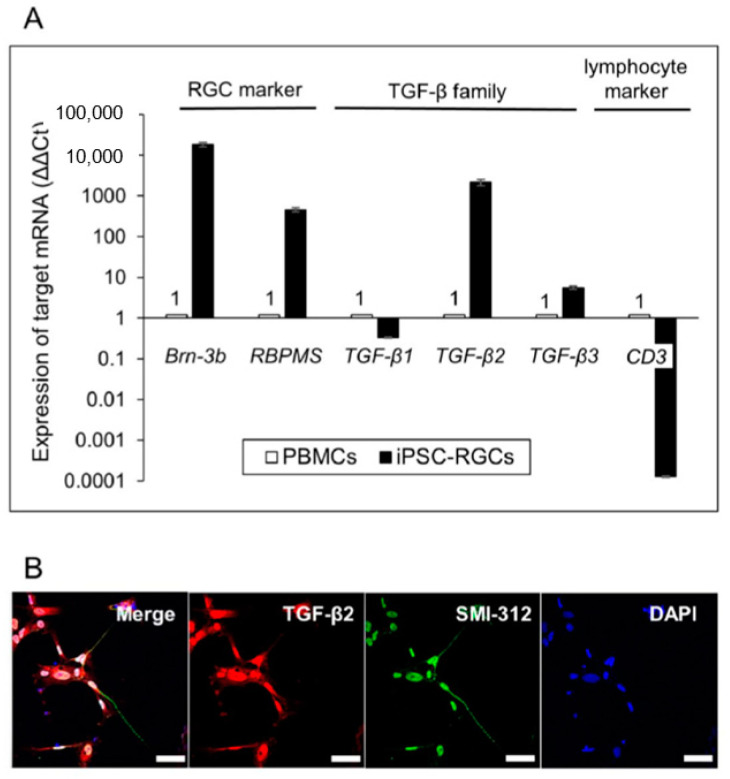
Detection of TGF-β in iPSC-RGCs. (**A**) Quantitative RT-PCR showed that iPSC-RGCs expressed *TGF-β2* and *-β3* mRNA, but *TGF-β1* mRNA was expressed at similar levels compared with control cells. On the other hand, iPSC-RGCs highly expressed *Brn-3b* and *RBPMS* mRNA compared with control cells. PBMCs (from the TLHD2 donor) were prepared as control cells. Results show the relative expression (∆∆Ct: control cells = 1) and represent the mean ± SD (*n* = 3). (**B**) ICC showed that iPSC-RGCs clearly expressed TGF-β2 (red), as well as SMI-312 (green), on their surface. No staining was seen with the isotype control (mouse, goat, and rabbit IgG). Cell nuclei were counterstained with DAPI. Scale bars, 50 µm.

**Figure 7 ijms-21-07831-f007:**
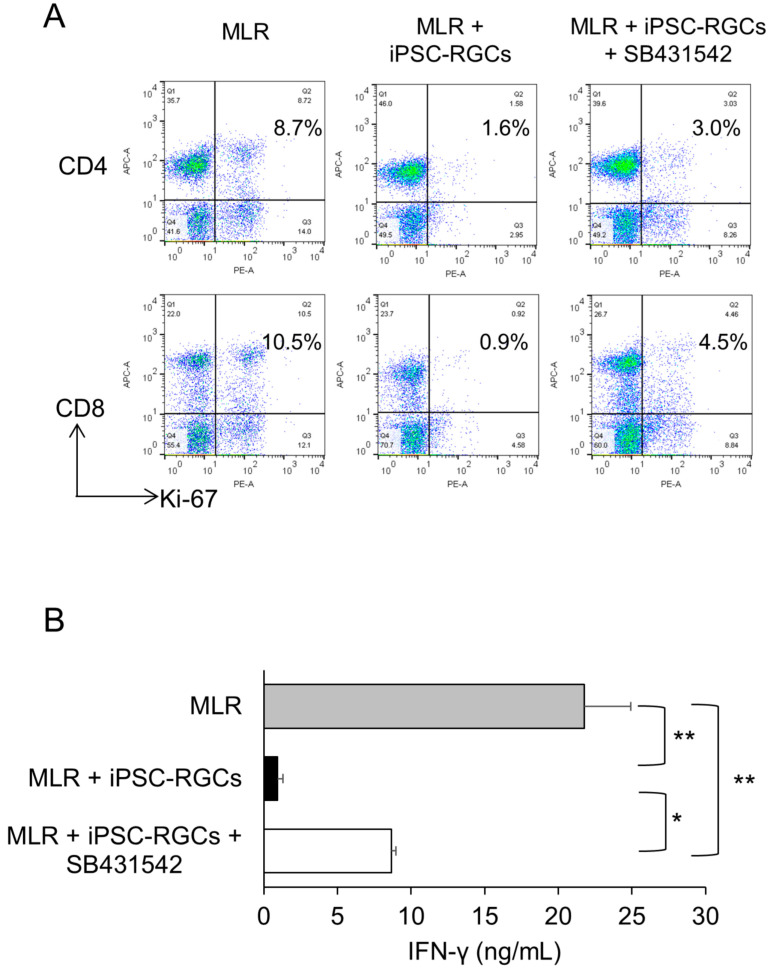
Capacity of TGF-β derived from iPSC-RGCs to suppress activation of T-cells. MLR cells were co-cultured with iPSC-RGCs (MLR:RGC ratio = 2:1) in the presence of the TGF-β inhibitor, SB431542. The activation of T-cells was analyzed with Ki-67 FACS and IFN-γ secretion with ELISA. (**A**) Representative data of Ki-67 FACS proliferation analysis. The percentage of double-positive cells that were co-cultured with iPSC-RGCs in the presence of SB431542 increased compared to that in co-cultures without SB431542. Numbers in the scatterplots indicate the percentage of double-positive cells for CD4 or CD8 and Ki-67. (**B**) Secretion of IFN-γ from MLR cells co-cultured with RGCs in the presence of SB431542 was significantly increased compared with no SB431542. The data represent the mean ± SD (*n* = 3). * *p* < 0.05, ** *p* < 0.01.

**Table 1 ijms-21-07831-t001:** Expression of genes related to immunogenicity and the immunosuppressive effect of iPSC-RGCs according to DNA microarray analysis.

Accession Number	Gene Description	Abbreviations		Signal Log2 Ratio	
			Signal in iPSC-RGCs	vs. iPSCs	vs. PBMCs
	***MHC Class I and Class II***				
NM_001242758	major histocompatibility complex, class I, A	HLA-A	56.74	−2.60	−7.91
ENST00000412585	major histocompatibility complex, class I, B	HLA-B	130.96	−2.00	−8.48
NM_001243042	major histocompatibility complex, class I, C	HLA-C	134.83	−3.20	−8.04
NM_001242525	major histocompatibility complex, class II, DP alpha 1	HLA-DPA1	64.49	−0.79	−8.87
NM_002121	major histocompatibility complex, class II, DP beta 1	HLA-DPB1	71.71	−3.08	−9.16
NM_002122	major histocompatibility complex, class II, DQ alpha 1	HLA-DQA1	30.59	−0.11	−9.89
OTTHUMT00000076179	major histocompatibility complex, class II, DQ alpha 2	HLA-DQA2	69.39	0.01	−5.88
NM_001243961	major histocompatibility complex, class II, DQ beta 1	HLA-DQB1	44.16	−0.86	−6.94
ENST00000411527	major histocompatibility complex, class II, DQ beta 2	HLA-DQB2	223.91	−0.52	−2.14
NM_019111	major histocompatibility complex, class II, DR alpha	HLA-DRA	34.59	−1.70	−11.94
NM_001243965	major histocompatibility complex, class II, DR beta 1	HLA-DRB1	44.85	−3.81	−10.23
NM_002125	major histocompatibility complex, class II, DR beta 5	HLA-DRB5	30.10	−0.98	−10.77
	***Co-stimulatory Molecules***				
NM_001250	CD40 molecule, TNF receptor superfamily member 5	CD40	29.43	−2.27	−5.70
NM_005191	CD80 molecule	CD80/B7-1	23.80	0.06	−4.26
NM_001206924	CD86 molecule	CD86/B7-2	12.31	−0.53	−6.30
NM_001267706	CD274 molecule	CD274/B7-H1/PD-L1	149.33	2.33	−4.57
NM_001024736	CD276 molecule	CD276/B7-H3	473.67	−0.96	2.16
NM_025239	programmed cell death 1 ligand 2	PDCD1LG2/PD-L2/B7-DC	26.47	1.13	−3.39
NM_001283050	inducible T-cell co-stimulator ligand	ICOSLG/B7-H2	61.92	0.00	0.00
NM_001253849	V-set domain containing T-cell activation inhibitor 1	VTCN1/B7-H4	39.01	0.42	0.49
NM_022153	chromosome 10 open reading frame 54	C10orf54/VISTA/B7-H5	67.94	0.00	−0.77
NM_003327	tumor necrosis factor receptor superfamily, member 4	TNFRSF4/OX40L	26.44	0.68	−2.68
NM_001252	CD70 molecule	CD70	13.67	0.19	−4.81
NM_001297605	tumor necrosis factor receptor superfamily, member 14	TNFRSF14/HVEM	75.79	0.51	−2.90
NM_005092	tumor necrosis factor (ligand) superfamily, member 18	TNFSF18/GITRL	9.20	−0.40	−0.55
NM_003811	tumor necrosis factor (ligand) superfamily, member 9	TNFSF9/4-1BBL/CD137L	240.76	−0.15	−0.15
	***Cytokines and Other Inhibitory Factors/Molecules***				
NM_000660	transforming growth factor beta 1	TGFB1	453.15	2.71	−3.85
NM_001135599	transforming growth factor beta 2	TGFB2	20563.82	3.85	7.86
NM_003239	transforming growth factor beta 3	TGFB3	30.27	0.22	0.16
NM_000639	Fas ligand (TNF superfamily, member 6)	FASLG	24.71	−0.24	−2.87
NM_004878	prostaglandin E synthase	PTGES	171.18	0.52	0.67
NM_001256335	prostaglandin E synthase 2	PTGES2	234.85	−0.57	−0.54
NM_000572	interleukin 10	IL10	19.01	−0.20	−2.77
NM_002164	indoleamine 2,3-dioxygenase 1	IDO1	37.34	−4.22	−8.23
NM_194294	indoleamine 2,3-dioxygenase 2	IDO2	43.79	−0.31	−0.31
NM_003381	vasoactive intestinal peptide	VIP	25.96	0.19	0.00
NM_002415	macrophage migration inhibitory factor	MIF	6137.52	−0.89	−0.93
NM_002389	CD46 molecule, complement regulatory protein	CD46	3081.43	−0.78	−0.35
NM_000574	CD55 molecule, decay accelerating factor for complement	CD55	2793.51	−1.37	−1.95
NM_000611	CD59 molecule, complement regulatory protein	CD59	10117.89	1.88	1.49
NM_003246	thrombospondin 1	THBS1/TSP-1	10138.58	5.59	7.24
NM_001048	somatostatin	SST	959.93	5.88	5.90
NM_000577	interleukin 1 receptor antagonist	IL1RN	17.64	0.26	−5.78
NM_001777	CD47 molecule	CD47	3346.94	1.88	−1.03

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
