# Peer review of "Capacity of Retinal Ganglion Cells Derived from Human Induced Pluripotent Stem Cells to Suppress T-Cells"

_ijms, 2020, doi:10.3390/ijms21217831_

Round 1

Reviewer 1 Report

  1. Authors must provide more informations on the background: Why is relevant the "Capacity of Retinal Ganglion Cells ... to suppress T Cells"?
  2. Authors must improve the paragraph 4.9 "Statistical analysis"; This section looks poor of contents emphasising well known informations: "p < 0.05 was considered statistically significant"; I'd like to know why authors have chosen "student's t-test" and "Tukey-Kramer test", they might explain the importance to use a parametric and a not parametric test in order to perform statistic supporting the own results.

Author Response

Reviewer 1

  1. Authors must provide more informations on the background: Why is relevant the "Capacity of Retinal Ganglion Cells ... to suppress T Cells"?

Response: We thank the reviewer for giving us this opportunity to clarify. The eye is known as an immune-privileged site, which has various mechanisms to prevent immune reactions. As one of the immune defense mechanisms, ocular resident cells such as retinal pigment epithelial cells (RPEs), corneal endothelial cells and iris pigment epithelial cells have been reported to have the capacity to suppress immune cells including T cells. However, it has not yet been reported whether RGCs have immunosuppressive properties.To assess the possibility of rejection of RGCs in transplantation therapy, we considered that it was important to ascertain whether iPSC-RGCs similarly have an immunosuppressive effect.

For clarification, we have added the following sentences:

‘Retinal pigment epithelial cells (RPEs), which is one of the constituent cells of the retina as well as RGCs, have been reported to possess immune suppression capacity [2-6]. Stem cell-derived RPE transplantation therapy has already been applied clinically [7-10]. On the other hand, to date, no reports have described the immunological characteristics of RGCs. We hypothesized that RGCs may also have the immunosuppressive properties like RPEs.’ (Revised manuscript, lines 37–41).

  1. Authors must improve the paragraph 4.9 "Statistical analysis"; This section looks poor of contents emphasising well known informations: "p < 0.05 was considered statistically significant"; I'd like to know why authors have chosen "student's t-test" and "Tukey-Kramer test", they might explain the importance to use a parametric and a not parametric test in order to perform statistic supporting the own results.

Response: Thank you for your comment.We analyzed with a Student's t-test for a comparison between the two groups that is the percentage of proliferating cells (Fig. 4B, C). Tukey-Kramer test was used for a comparison of more than three groups that is IFN-gconcentrations (Fig. 4D, Fig. 5, Fig. 7B), as a multiple comparison. However, we have noticed that comparison among three groups are not necessary, so instead we have re-compared each two groups by Student’s t-test. Since all experiments for statistical analysis were performed in experimental triplicate, we analyzed with parametric analysis. Thus, we have revised the sentences as follows:

‘All in vitroexperiments were repeated three times. Data are presented as mean ±standard deviation (SD) of the experimental triplicate. All statistical analysis was performed by Student’s t-test using JMP pro 14.0.0 (SAS Institute Inc., Cary, NC, USA).’ (Revised manuscript, lines 472–477).

Reviewer 2 Report

The authors aimed to characterize the immune properties of iPSC-RGCs and the study is well designed and the results are well described. However, there are several concerns that authors should consider.

In the introduction (and in the abstract) the authors argue that “Characterization of RGC immunology is important for predicting immune rejection after RGC transplantation.” At this point, it seems that this is a gap that is missing in the success of RGC transplantation in the future. It means that the authors want to characterize the RGC immune response in order to predict immune rejection in humans. However, this issue is no longer raised throughout the manuscript. Is it possible, with the methods and the results described in this manuscript, to predict if the cells will be rejected or not? How this study contributes to predict immune attacks after RGCs transplantation? This issue should be raised in the introduction and clearly discussed in the discussion part of the manuscript.

In the discussion, the authors start by describing the results again, instead of discussing them comparing with the literature. From line 258 until line 297 the authors only describe the results and most of the references are from published work from co-authors of this manuscript. The authors should re-organize the discussion in order to clearly discuss their results supported by the literature.

Some minor concerns:

Line 37 (Introduction): “via their axons which is the optic nerve”. The optic nerve is not only formed by the axons of RGCs, the way that is written it seems that the RGCs axons are the optic nerve. The authors should consider rewording this idea.

Line 40 (Introduction): “RGCs are part of the central nervous system”. Retina is central nervous system not only RGCs. The authors should consider rewording this idea.

Author Response

Reviewer 2

The authors aimed to characterize the immune properties of iPSC-RGCs and the study is well designed and the results are well described. However, there are several concerns that authors should consider.

In the introduction (and in the abstract) the authors argue that “Characterization of RGC immunology is important for predicting immune rejection after RGC transplantation.” At this point, it seems that this is a gap that is missing in the success of RGC transplantation in the future. It means that the authors want to characterize the RGC immune response in order to predict immune rejection in humans. However, this issue is no longer raised throughout the manuscript. Is it possible, with the methods and the results described in this manuscript, to predict if the cells will be rejected or not? How this study contributes to predict immune attacks after RGCs transplantation? This issue should be raised in the introduction and clearly discussed in the discussion part of the manuscript.

Response: We thank the reviewer for this comment.We do not believe that rejection can be predicted solely on the basis of this study. Because we consider that the accumulation of findings, such as this study, will help to assess the possibility of rejection in the future,we performed the present study as one of the basic insights for it. We have corrected the misleading expression and have added the explanations. The rejection is the activation of host immune cells such as T cells against transplanted cells by the recognition of the transplanted cells as foreign.It is known that the eye is a unique environment, an immune-privileged site, with a variety of mechanisms to prevent immune responses. One mechanism for this is that the retinal pigment epithelial cells (RPEs), corneal endothelial cells and iris pigment epithelial cells have been reported to be capable of inhibiting the activation of immune cells in vitro. It has not been known whether RGCs also have the capacity to suppress immune cells. Therefore, we performed the mixed lymphocyte reaction (MLR) assay, which is extensively used to assess immune rejection in vitro, to evaluate whether RGCs also have immunosuppressive effects. Furthermore, since high expression of immunogenic molecules such as HLA class I, class II, CD80 and CD86 makes the transplants more likely to be recognized as foreign by host immune cells, in this study we showed there were low or no expression of these molecules on RGCs. The purpose of this study was to examine the immune properties (the immunogenicity and the capacity to suppress T cells) of RGCs.

We have revised some sentences as follows:

‘to evaluate the possibility of rejection’ (Revised manuscript, line 21),

‘Characterization of RGC immunology is important for assessing the possibility ofimmune rejection…’ (Revised manuscript, lines 62–63).

In addition, we have revised the following sentences in the Introduction section:

‘Retinal pigment epithelial cells (RPEs), which is one of the constituent cells of the retina as well as RGCs, have been reported to possess immune suppression capacity [2-6]. Stem cell-derived RPE transplantation therapy has already been applied clinically [7-10]. On the other hand, to date, no reports have described the immunological characteristics of RGCs. We hypothesized that RGCs may also have the immunosuppressive properties like RPEs.’(Revised manuscript, lines 37–41),

‘Especially in allogenic transplantation, rejection is a critical concern. In general, CD4-positive and CD8-positive T cells are largely responsible for rejection [22-24]. Expression of human leukocyte antigen (HLA) and CD80/CD86 co-stimulatory molecules is a major trigger of immune response, and high expression these molecules increases the risk of rejection [25-27]. T-cell suppressive capacity and low expression of HLA molecules are associated with the low incidence of rejection.’ (Revised manuscript, lines 57–62),

‘In the present study, we evaluated the immunogenicity of iPSC-RGCs such as expression of HLA molecules, and CD80, CD86 and CD274 co-stimulatory molecules. We also investigated whether iPSC-RGCs have a T cell suppression capacity using the mixed lymphocyte reaction (MLR) assay, that is extensively used for assessing immune rejection in vitro[28-32]. To elucidate the immunosuppressive property and the immunogenicity of human iPSC-derived RGCs (iPSC-RGCs) against lymphocytes, we used an in vitromodel with iPSC-RGCs derived from healthy donors co-cultured with active lymphocytes isolated from the peripheral blood of healthy subjects.’ (Revised manuscript, lines 65–76).

We have also revised the following sentences in the Discussion section:

‘Based on our results mentioned above,…’(Revised manuscript, line 334),

‘Cytotoxicity generated by CD8-positive T cells upon recognition of HLA class I expressed on transplanted cells has been reported to be the main cause of rejection. CD4-positive helper T cells activated by antigen presentation via HLA class II also promote rejection [22-24].’ (Revised manuscript, lines 337–330),

 ‘The vitreous contains a variety of anti-inflammatory factors such as TGF-b, vasoactive intestinal peptide, alpha-melanocyte stimulating hormone (a-MSH), somatostatin, and substance P [50]. Moreover, the vitreous has a mechanism to induce systemic immune tolerance by recognizing antigens and raising regulatory T cells with anti-inflammatory effects. When antigens are injected into the vitreous, antigen-presenting cells in the vitreous or in the retina migrate to the spleen for antigen presentation, and activate antigen-specific regulatory T cells to suppress the immune response in the vitreous [50, 52, 53].’ (Revised manuscript, lines 342–350),

‘Immune rejection occurs after allogeneic transplantation of iPSC-RPEs in vivo, despite the fact that the transplant iPSC-RPEs are capable of inhibiting T cell activity and the transplantation site, the sub-retinal space, is an environment less prone to inflammation as well as the vitreous [54-56].’ (Revised manuscript, lines 351–356),

‘In animal models of RGC transplantation, in which mouse RGCs were transplanted into rat vitreous, transplanted cells survived and no obvious signs of rejection were observed at 3 weeks after transplantation without the use of immunosuppressive drugs [17]. This suggests that the risk of rejection associated with RGC transplantation is relatively low. The next step to predict rejection based on the results of this study would be to evaluate the immune rejection in animal transplants using iPSC-RGCs. We hope that our study will support the development of strategies against rejection for RGC transplantation therapy that is becoming nearing reality.’ (Revised manuscript, lines 368–375).

In the discussion, the authors start by describing the results again, instead of discussing them comparing with the literature. From line 258 until line 297 the authors only describe the results and most of the references are from published work from co-authors of this manuscript. The authors should re-organize the discussion in order to clearly discuss their results supported by the literature.

Response: We thank the reviewer for this suggestion. We agree and have incorporated this suggestion to our paper. We have re-organized the discussion section. We couldn't find the literature that mentions TGF-b-mediated T-cell suppression of human corneal endothelial cells and human iris pigment epithelial cells published other than our co-authors.

We have revised the following sentences in the Discussion section:

‘In this study, we found that iPSC-RGCs have poor expression of HLA class I and no expression of HLA class II, CD80 and CD86 co-stimulatory molecules. When HLA class I of transplanted cells is recognized by immune cells, CD8-positive cytotoxic T cells are activated to eliminate the grafts [22-24]. HLA class II, CD80 and CD86 co-stimulatory molecules are expressed on the antigen-presenting cells [25, 31, 38]. Our findings imply that iPSC-RGCs are less likely to activate CD8-positive cytotoxic T cells and do not have an antigen presenting function.

iPSC-RGCs expressed CD274 co-stimulatory molecules, which is a ligand of PD-1 expressed on T cells [23]. PD-1 pathway inhibits the activation of CD4-positive and CD8-positive T cells in vitro [41]. In allogenic heart transplants in mice in vivo, graft survival was prolonged by promoting PD-1 pathway [42]. This suggests that CD274 expression on iPSC-RGCs might have an advantage in preventing rejection.

We showed that iPSC-RGCs suppressed the immune activity of T cells via TGF-b. Human corneal endothelial cells, iris pigment epithelial cells and stem cell-derived RPEs have been reported to suppress T cell activation via TGF-b[4, 6, 43, 44]. In particular, it has been suggested that the induction of regulatory T cells by TGF-bis involved in the mechanism of inhibition by human corneal endothelial cells and iPSC-RPEs [4, 44]. In the present study, the involvement of TGF-bhas been suggested and a similar mechanism can be inferred.’ (Revised manuscript, lines 275–321).

We have added the following sentence:

‘In human RPEs, some factors other than TGF-b, such as prostaglandin E2 and soluble CD54, have also been reported to be involved in suppression of T cell activity [3].’ (Revised manuscript, lines 323–325).

Some minor concerns:

Line 37 (Introduction): “via their axons which is the optic nerve”. The optic nerve is not only formed by the axons of RGCs, the way that is written it seems that the RGCs axons are the optic nerve. The authors should consider rewording this idea.

Response: We agree with your comments. We have revised as follows:

‘Retinal ganglion cells (RGCs), which are located in the innermost layer of the retina, are responsible for collecting optical information that reaches the retina, and transmitting it to the brain via the optic nerve[1].’ (Revised manuscript, lines 35–37).

Line 40 (Introduction): “RGCs are part of the central nervous system”. Retina is central nervous system not only RGCs. The authors should consider rewording this idea.

Response: Thank you for your comment. We have revised it as follows:

‘Like other neurons of the central nervous system, RGCs do not readily self-replicate,…’ (Revised manuscript, lines 45–46).

We really appreciate the helpful comments.

Round 2

Reviewer 2 Report

I am satisfied by the authors' response and the authors improve their discussion of the results.